# Computed tomography evaluation of risk factors for an undesirable buccal split during sagittal split ramus osteotomy

Yasuyuki Fujii[1]*, Ayano Hatori[1], Miwa Horiuchi[1], Tomoko Sugiyama-Tamura[1], Hayato Hamada[1], Risa Sugisaki[1], Yuki Kanno[2], Marika Sato[1], Michihide Kono[1], On Hasegawa[1], Yoko Kawase-Koga[2], Daichi Chikazu[1]

1 Department of Oral and Maxillofacial Surgery, Tokyo Medical University, Shinjuku-ku, Tokyo, Japan,
2 Department of Oral and Maxillofacial Surgery, Tokyo Women's Medical University, Shinjuku-ku, Tokyo, Japan

* yfujii@tokyo-med.ac.jp

**Data Availability Statement:** All relevant data are within the paper and its Supporting Information files.

## Abstract

Sagittal split ramus osteotomy (SSRO) sometimes induces an irregular split pattern referred to as a bad split. We investigated the risk factors for bad splits in the buccal plate of the ramus during SSRO. Ramus morphology and bad splits in the buccal plate of the ramus were assessed using preoperative and postoperative computed tomography images. Of the 53 rami analyzed, 45 had a successful split, and 8 had a bad split in the buccal plate. Horizontal images at the height of the mandibular foramen showed that there were significant differences in the ratio of the forward thickness to the backward thickness of the ramus between patients with a successful split and those with a bad split. In addition, the distal region of the cortical bone tended to be thicker and the curve of the lateral region of the cortical bone tended to be smaller in the bad split group than in the good split group. These results indicated that a ramus shape in which the width becomes thinner towards the back frequently induces bad splits in the buccal plate of the ramus during SSRO, and more attention should be paid to patients who have rami of these shapes in future surgeries.

## Introduction

Sagittal split ramus osteotomy (SSRO) is a common surgical method for the correction of mandibular prognathism. Compared with intraoral vertical ramus osteotomy, which is another common method for orthognathic surgery, SSRO requires a shorter period of inter-maxillary fixation because of the large area of bony contact and the rigid fixation. However, SSRO sometimes induces an irregular split pattern referred to as a bad split. The incidence of bad splits in SSRO from 21 studies varied between 0.21% and 22.72% [1]. A bad split in sagittal split ramus osteotomy (SSRO) is defined as an unfavorable fracture pattern of the mandible during osteotomy or splitting [2,3]. Bad splits during SSRO have been reported to occur in the buccal plate of the proximal segment, lingual plate of the distal segment, coronoid process, and condylar neck [4]. Among them, the buccal plate of the proximal segment is the most frequent

**Funding:** Initials of the authors who received each award: YF Grant numbers awarded to each author: (No. 21K17098). The full name of each funder: Japan Society for the Promotion of Science (JSPS) KAKENHI Grant. URL of each funder website: https://www.jsps.go.jp/english/index.html Did the sponsors or funders play any role in the study design, data collection and analysis, decision to publish, or preparation of the manuscript?: No.). Initials of the authors who received each award: DC Grant numbers awarded to each author: (No. 22K10230). The full name of each funder: Japan Society for the Promotion of Science (JSPS) KAKENHI Grant. URL of each funder website: https://www.jsps.go.jp/english/index.html Did the sponsors or funders play any role in the study design, data collection and analysis, decision to publish, or preparation of the manuscript?: No.

**Competing interests:** No competing interests exist.

site of fracture. Buccal plate fractures in the anterior site may heal just by removing the small segment. However, it is difficult to treat a bad split in the ramus. An undesirable buccal split in the ramus results in bony interference between the proximal and distal segments during set-back movement, or results in attachment of the condyle to the distal segment. Salvage surgical approaches for buccal fractures in the ramus are sometimes challenging. Previous studies have reported risk factors for bad splits during SSRO [5–8], but there is controversy regarding some of the risk factors. It is still unclear whether older age of the patients, the presence of the third molar, and mandibular anatomy increases the risk of bad splits during SSRO.

Little is known about whether the anatomical features of the ramus affect the risk of buccal fractures in the proximal segment. Therefore, the aim of this study was to investigate the risk factors for bad splits in the buccal plate of the ramus during SSRO by analyzing the anatomy of the ramus using computed tomography (CT).

## Materials and methods

### Patients and surgical treatments

The Ethics Committee of the Faculty of Medicine, Tokyo Medical University reviewed and approved the study design (study approval number: T2021-0061). Informed consent was obtained from all individual participants included in the study. A total of 27 Japanese patients (13 men and 14 women) who were treated by SSRO at Tokyo Medical University Hospital between January and December 2020 were analyzed. The age of the patients at the time of orthognathic surgery ranged from 17 to 51 years (mean: 25.5 years; median: 24.0 years). SSRO procedures were performed basically according to the Hunsuck-Epker modification, which is known as short lingual osteotomy (SLO) [9,10]. Patients who underwent extraction of the lower third molars or advanced movement of the proximal segment during SSRO were excluded. Patients who underwent advanced movement of the proximal segment during SSRO were also excluded because SSRO procedures in these patients were basically performed according to the Obwegeser-Dal Pont method. Features of the jaw deformities were mandibular prognathism with/without maxillary retrusion and/or facial asymmetry.

A modified version of SLO that has been described in a previous study [11], in which an ultrasonic bone-cutting device is used to determine the posterior osteotomy boundary, was performed in all cases. Briefly, after incision of the mucosa of the anterior border of the ramus, the periosteum was detached from the inferior border to the posterior border of the lateral aspect of the ramus, and then was also detached from the medial aspect extending horizontally in a posterior direction from between the sigmoid notch and mandibular foramen up to the posterior border of the ramus. Horizontal osteotomy was performed above the lingula of the mandible using a Lindemann bur. The cortical bone was cut from the lateral aspect of the anterior mandibular body to the medial aspect of the ramus using a reciprocating saw. Vertical osteotomy of about 10 mm was then conducted from the posterior edge of the horizontal osteotomy line toward the inferior border, using an ultrasonic bone-cutting device (SONO-PET; Stryker Corporation) to split the sagittal surface of the ramus.

To ensure that all surgeries were performed using the same surgical technique, the chairman of our department, who has been performing craniofacial surgeries for more than 30 years and specializes in orthognathic surgery, participated and supervised all surgeries. The same instruments were used in all surgeries in this study. Nine surgeons, including the chairman, performed all operations. The 9 surgeons were all members of the Japanese Society of Oral and Maxillofacial Surgeons, who have been performing craniofacial surgery for 9 or more years.

## CT analysis of rami anatomy and buccal fracture of the rami

CT analyses were performed about 1 month before the surgery, and 4 or 5 days after the surgery, using a Revolution CT device (GE Healthcare) at Tokyo Medical University Hospital (tube voltage: 120 kV; tube current: auto mA; rotation time: 0.5 s/rotation; slice width: 2.5 mm; slice interval: 0.625 mm; field of view: 23 cm; reconstruction kernel: bone; scan pitch: 0.561).

Horizontal images were obtained at the height of the lingula of the mandible (Image A, Fig 1A) and the mandibular foramen (Image B, Fig 1B), and the landmarks for measurement were manually identified (Fig 1C and 1D); in Image A, the forward point (A), the backward point (B), the intersection point of the perpendicular bisector of line AB and the lateral surface of the buccal cortical bone (C), in Image B, the medial point (D), the point on the medial surface 5 mm behind point D (E), the point of tangency of the mandibular foramen and point D (F), the center of the mandibular foramen (G), the point of tangency of the lateral surface of the lingual cortical bone and point F (H), the backward point (I), the point on the medial surface 5 mm in front of point I (J), the point on the lateral surface 5 mm in front of point I (K), the forward point (L), the point on the lateral surface 5 mm behind point L (M), and the intersection point of the perpendicular bisector of line IL and the lateral surface of the buccal cortical bone (N). Angle ACB as the curve of the medial region of the cortical bone and the thickness of the distal region of the cortical bone were measured in Image A. Angle DFH as the curve of the medial region of the cortical bone, angle LNI as the curve of the lateral region of the cortical bone, the distance of EM as the forward thickness of the ramus, the distance of JK as the backward thickness of the ramus, the distance of GI, and the thickness of the distal region of the cortical bone were measured in Image B. Whether or not the mandibular foramen was in direct contact with the buccal cortical bone was also evaluated in Image B. Bad splits in the buccal plate of the ramus were evaluated in Image A or B after surgery, when a split line appeared on the buccal cortical bone 3 mm or more lateral from point I (Fig 1E). These measurements were conducted 3 times by 2 experienced oral surgeons, and the mean values were analyzed. Measurements were performed once a month, and 2 surgeons performed the measurements individually.

## Bone quality analysis by measurement of Hounsfield Units (HUs)

To evaluate bone quality, HU values in the ramus were analyzed using Simplant software (Dentsply Sirona). HUs were measured in 2 rectangular areas in Images A and B. The rectangle was located 3 mm (anterior area) or 8 mm (posterior area) in front of the backward point (Fig 4A and 4B). The width of the rectangle was set at 1.5 mm, and the length was defined as the distance from the buccal surface to the lingual surface. Anterior and posterior HUs were defined as the average HU in Images A and B, respectively. These measurements were conducted 3 times by 2 oral surgeons, and the mean values were analyzed (S1C–S1F Fig in S1 Raw data). Measurements were performed once a month, and 2 surgeons performed the measurements individually.

## Statistical analysis

Statistical analysis was performed by the Student *t*-test or the Fisher exact test using Prism 8 software (GraphPad Software). A *p*-value of less than 0.05 was considered to indicate a statistically significant difference between 2 groups. Intra-rater reliability was calculated using the intraclass correlation coefficient (ICC) by SPSS 24.0 software (IBM).

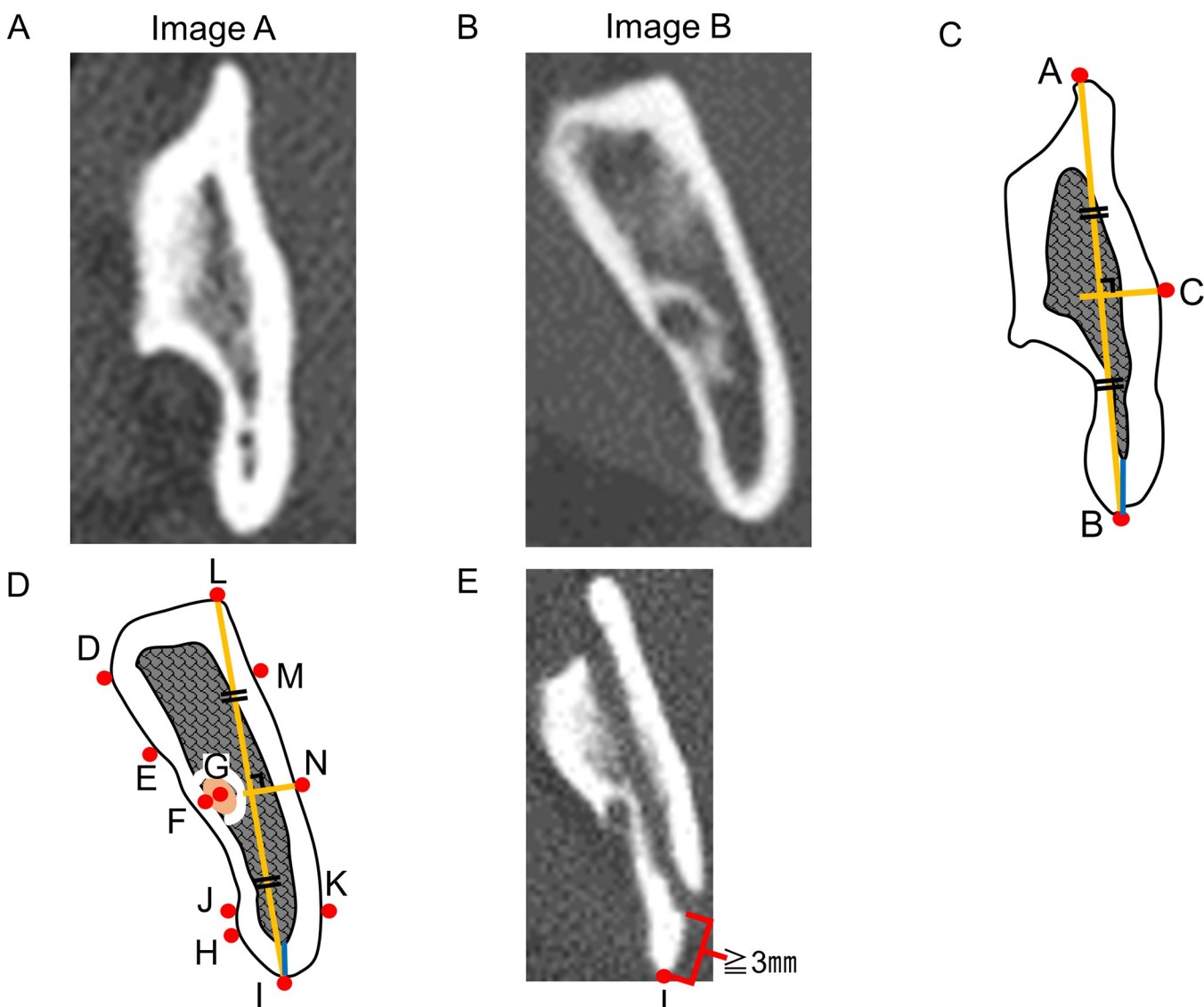

**Fig 1. Measurement of mandibular morphology in the ramus and evaluation of bad splits during SSRO.** Horizontal images at the height of the lingula of the mandible **(Image A, A)** and the mandibular foramen **(Image B, B)** were obtained on preoperative CT, and the landmarks for measurement were manually identified in Image A **(C)** and Image B **(D)**; in Image A, the forward point (point A), the backward point (point B), the intersection point of the perpendicular bisector of line AB and the lateral surface of the buccal cortical bone (point C), in Image B, the medial point (point D), the point on the medial surface 5 mm behind point D (point E), the point of tangency of the mandibular foramen and point D (point F), the center of the mandibular foramen (point G), the point of tangency of the lateral surface of the lingual cortical bone and point F (point H), the backward point (point I), the point on the medial surface 5 mm in front of point I (point J), the point on the lateral surface 5 mm in front of point I (point K), the forward point (point L), the point on the lateral surface 5 mm behind point L (point M), and the intersection point of the perpendicular bisector of line IL and the lateral surface of the buccal cortical bone (point N). The thickness of the distal region of the cortical bone was measured in both Images A and B (blue lines in **(C)** and **(D)**, respectively). Bad splits in the buccal plate of the ramus were evaluated in Image A or B after surgery, when the split line appeared on the buccal cortical bone 3 mm or more lateral from point I **(E)**.

## Results

One rami which was operated for advanced movement of the proximal segment was excluded. Of the 53 rami analyzed, 45 had a successful split (good split group: 84.9%), and 8 had a bad split in the buccal plate of the ramus (bad split group: 15.1%, Fig 2). There was no significant difference in the age of the patients between the good split and the bad split groups. Bad splits occurred more frequently in male patients (87.5%) than in female patients. In Image A, there

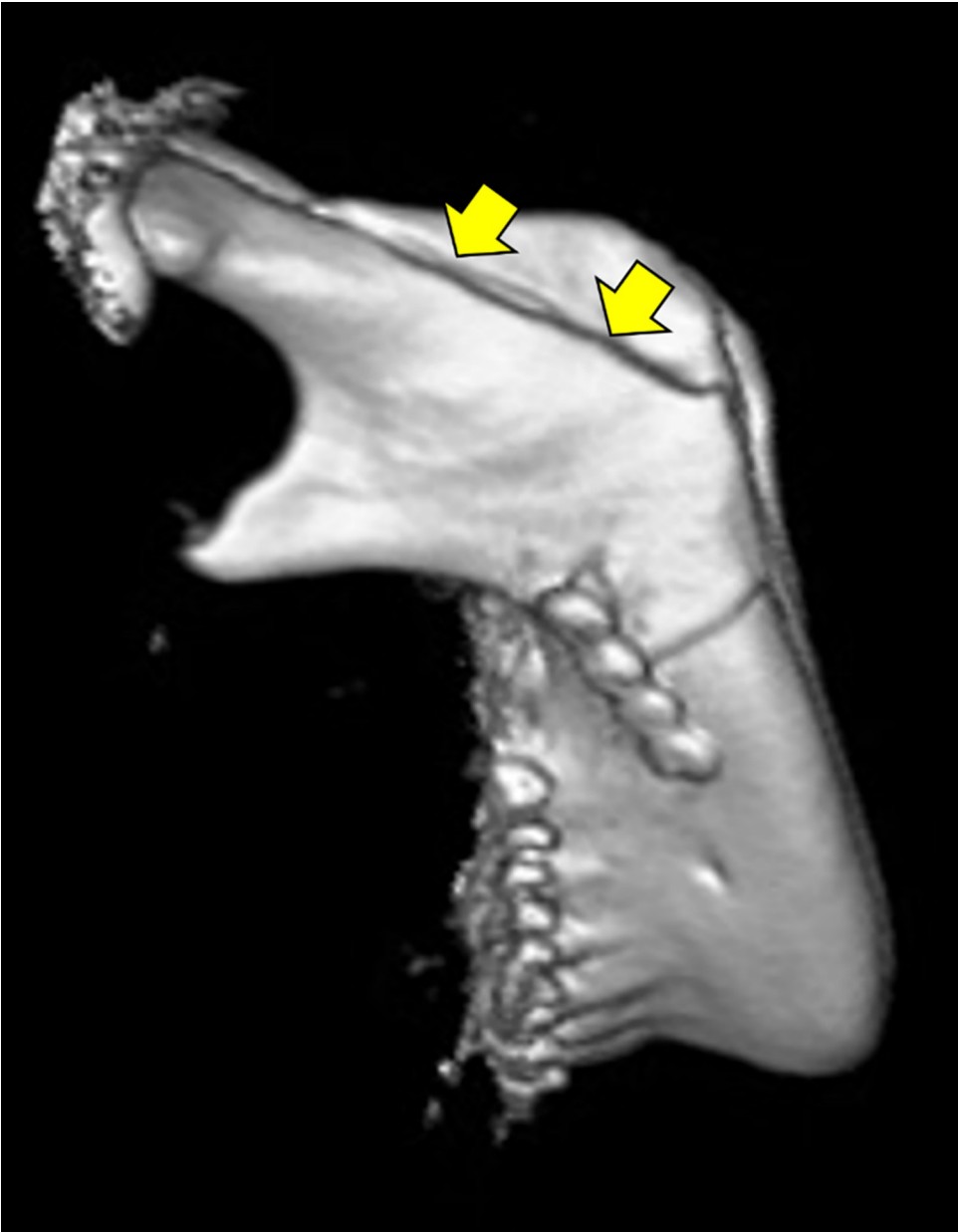

**Fig 2. Representative three-dimensional image of an undesirable buccal split during SSRO.** The yellow arrows indicate a bad split in the buccal plate of the ramus.

were no significant differences in angle ACB and the thickness of the distal region of the cortical bone between the good split group and the bad split group (Fig 3A and 3B). On the other hand, there was a statistically significant difference in the ratio of EM to JK between the two groups ($p = 0.037$) (Fig 3C). In addition, the distal region of the cortical bone in the bad split group tended to be thicker ($p = 0.099$) (Fig 3D), and angle LNI in the bad split group tended to be smaller than in the good split group ($p = 0.087$) (Fig 3E), although the difference was not statistically significant. There were also no significant differences in angle DFH and the distance of GI between the two groups (Fig 3F and 3G). In four patients of the good split group and one patient of the bad split group, the mandibular foramen was in direct contact with the

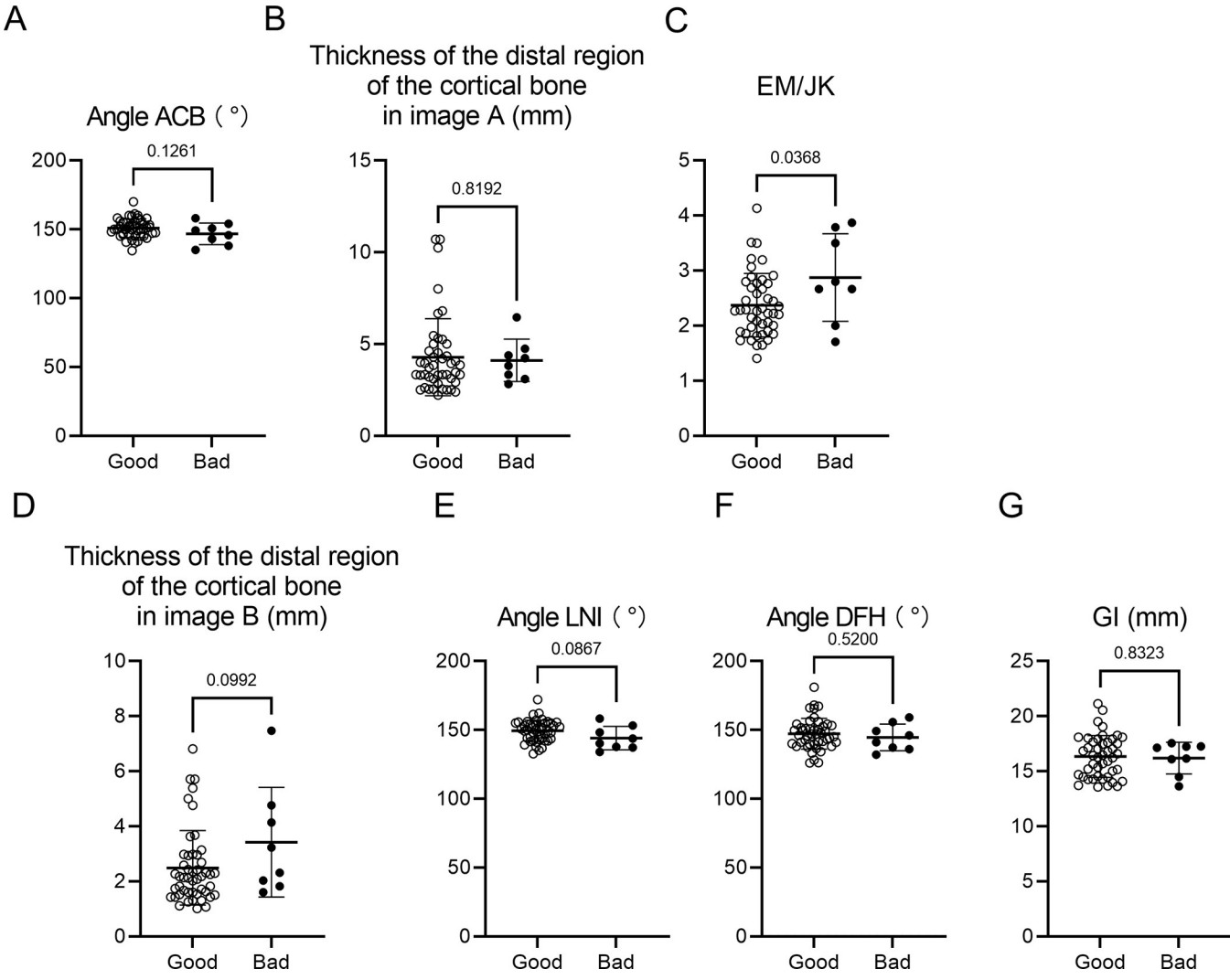

**Fig 3. Analysis of mandibular morphology in the ramus and evaluation of bad splits occurring during SSRO.** Comparisons between the good split and the bad split group regarding angle ACB (**A**), thickness of the distal region of the cortical bone in Image A (**B**), the ratio of EM to JK (**C**), thickness of the distal region of the cortical bone in Image B (**D**), angle LNI (**E**), angle DFH (**F**), and distance of GI (**G**). Data are presented as mean ± SD.

buccal cortical bone, although there was no significant difference in the frequency between the two groups (Table 1; $p = 0.574$). The ICC values for the measurements of the CT images were from 0.966 to 0.998.

The anterior HUs of the bad split group tended to be lower than those of the good split group ($p = 0.0717$) (Fig 4C); however, the mean posterior HUs of the 2 groups were almost the

**Table 1. Comparison of the location of the mandibular foramen between the good split group and bad split group in Image B.**

| Group | Direct contact with buccal cortical bone | No contact with buccal cortical bone | Total |
|---|---|---|---|
| Good split | 4 (8.333) | 44 (91.667) | 48 (100) |
| Bad split | 1 (12.5) | 7 (87.5) | 8 (100) |

Data are shown as number of patients (%).

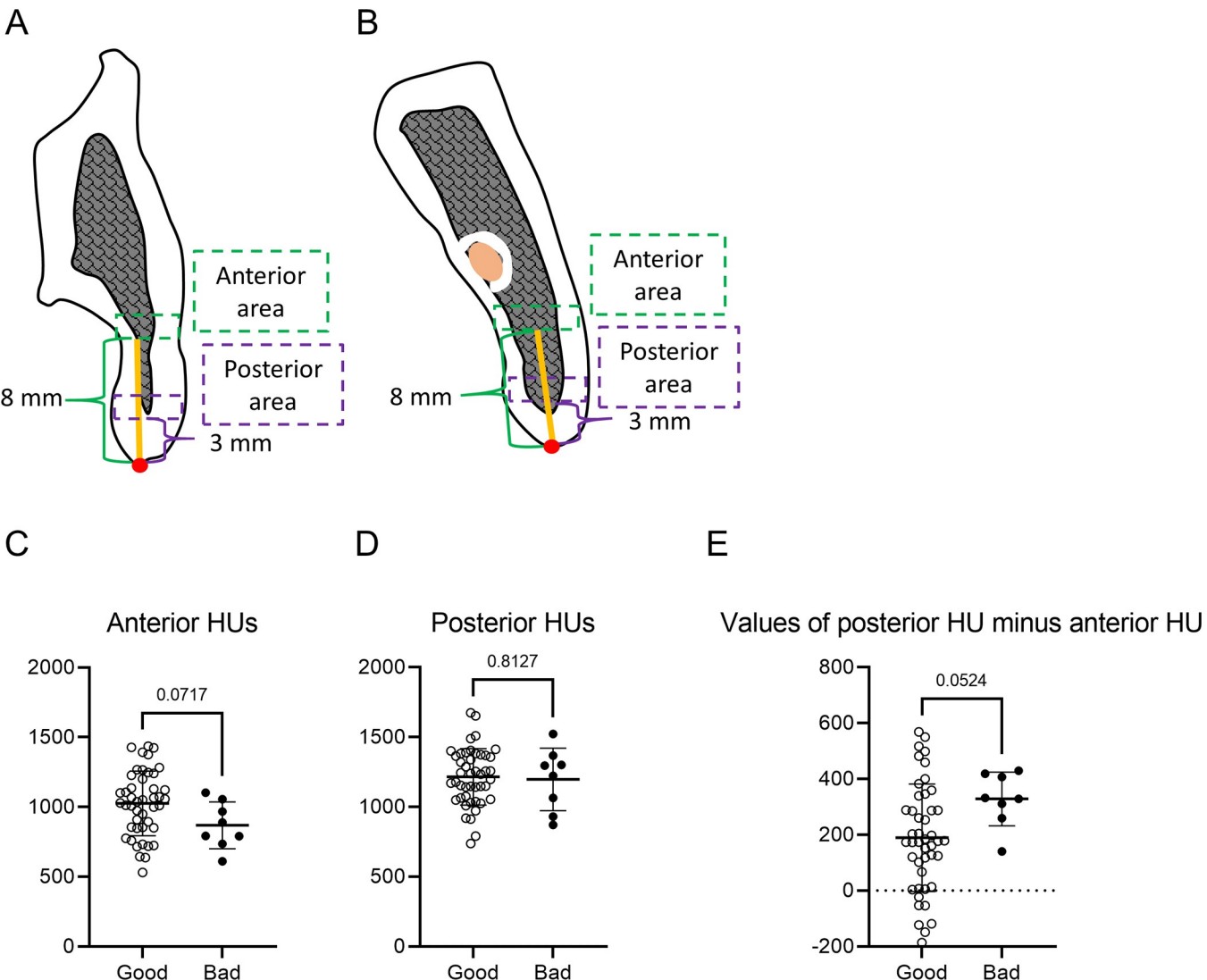

**Fig 4. Analysis of HUs in the ramus.** HUs were measured in 2 rectangular areas at the height of the lingula of the mandible (**A**) and the mandibular foramen (**B**). The rectangle was located 3 mm (anterior area) or 8 mm (posterior area) in front of the backward point. The rectangle width was set at 1.5 mm, and the length was defined as the distance from the buccal surface to the lingual surface. Anterior HUs (**C**) and posterior HUs (**D**) of the 2 groups, measured from Images A and B, respectively. (**E**) Values of posterior HU minus anterior HU of the 2 groups. Data are presented as mean ± SD.

same (good split group: 1,216; bad split group: 1,197) (Fig 4D). Values of posterior HU minus anterior HU of the bad split group also tended to be higher than those of the good split group ($p = 0.0524$) (Fig 4E). In addition, the posterior HUs were all higher than the anterior HUs in the bad split group. The ICC value for the measurements of bone quality analysis was 0.972.

## Discussion

Bad split is a major complication in SSRO. Among the cases of bad splits, the most frequent site is the buccal plate of the proximal segment [12]. Chrcanovic BR. Et al reported that the incidence of bad splits in SSRO from 21 studies varied between 0.21% and 22.72% [1]. The other reported incidence of bad splits in SSRO ranges from 0.5% to 5.5% per site [13,14]. In the present study, the rate of bad splits in the buccal plate might be higher than that of previous

studies. This difference may be owing to the definition of a bad split. We defined a bad split as a buccal split, which is the appearance of a split line on the buccal cortical bone 3 mm or more lateral from point I. A split line near the posterior edge of the buccal cortical bone might not result in any problems during surgery.

Risk factors of a bad split in SSRO, such as older age, a thick mandible, surgical technique, horizontal osteotomy line, and presence of the lower third molars have previously been investigated [5–8,15,16]. However, the results remain controversial. For example, a previous study demonstrated that the presence of mandibular third molars increased the risk of a bad split [17]. However, other studies reported that the presence of mandibular third molars do not increase the risk of bad splits [6,18]. Surgical procedures might also have an effect on the incidence of bad splits during SSRO. Zeynalzadeh F. et al reported that the incidence of unfavorable fractures Hunsuck technique is less compared to the Dal Pont osteotomy technique [19].

Mandibular anatomy also affects the risk of bad splits during SSRO. Previous studies suggested that patients with a shorter ramus and thinner buccolingual alveolar region distal to the second molar have a higher risk of bad splits than other patients [7,20]. Moreover, in a study by Aarabi et al., patients with a shorter ramus and buccolingually thin mandible were found to be more susceptible to fracture during sagittal osteotomy than other patients [21]. In the present study, we analyzed mandibular anatomy using preoperative CT to determine whether the shape of the ramus affects the risk of bad splits in the buccal plate of the proximal segment. Compared with the good split group, the ratio of EM to JK was larger in the bad split group. This result indicates that a ramus shape in which the width becomes thinner from the front to the back increases the risk of bad splits in the buccal plate. Angle LNI, which was regarded as the curve of the lateral region of the cortical bone at the height of the mandibular foramen, in the bad split group tended to be smaller than that in the good split group, although the difference was not statistically significant. The split may be caused by pressure applied on the distal aspect of the posterior ramus during SSRO in patients with such a mandibular shape. The distal region of the cortical bone also tended to be thicker in the bad split group. This shape may result in force being applied to the medial or distal aspect of the posterior ramus in the front of the distal region of the cortical bone, because it is difficult to split the thick cortical bone. In addition, our results also showed that posterior HUs were all higher than anterior HUs in the bad split group, although there was no significant difference in the values of posterior HU minus anterior HU (Fig 4E). Therefore, a higher posterior HU might induce short splits in not only the lingual side but also in the buccal side. Further studies are needed to obtain more evidence regarding this point.

There were no differences in ramus shape between the two groups in the analysis of horizontal images at the height of the lingula of the mandible. It was difficult to identify the points of measurement, such as the medial point and the center of the mandibular foramen, because the lingula of the mandible had various shapes. One limitation of the present study is that only 1 anatomical factor was indicated as a risk factor. We selected 2 areas at the height of the lingula of the mandible and the mandibular foramen because these 2 points could easily be identified in the horizontal image. It was sometimes difficult to perform measurements owing to metal artifacts, and to identify anatomical points for measurement in the other axial images. Anatomical analyses using horizontal images (either coronal or sagittal) at other heights of the ramus, or using three-dimensional images may reveal more details regarding the association between mandibular anatomy and the occurrence of bad splits in the buccal plate of the proximal segment during SSRO. Another limitation is that the patients in this study underwent operations performed by different surgeons with various surgical experiences, which might have resulted in inter-operator bias in evaluating the outcomes. To minimize this bias, an expert in orthognathic surgery participated in and supervised all surgeries, to ensure the use of

the same surgical technique and the same instruments. In the present study, 9 surgeons performed the surgeries. This is a rather large number of surgeons for performing surgeries on only 27 patients, which can increase the variability substantially and reduce the statistical power. Therefore, further studies involving a larger number of samples, and a multicenter study analyzing other risk factors, such as surgical technique, are needed to obtain more evidence to support our present results.

## Conclusions

Patients with a ramus shape in which the width becomes thinner from the front to the back, a cortical bone that is thick in the distal region, or a lateral cortical bone that has a sharp curve at the height of the mandibular foramen are more susceptible to bone fracture during SSRO than other patients.

## Supporting information

**S1 Raw data.**
(XLSX)

## Acknowledgments

We would like to thank all staff members of Tokyo Medical University hospital for their assistance during data collection. We acknowledge all study participants for their willingness to take part in the study.

## Author Contributions

**Data curation:** Hayato Hamada, Risa Sugisaki, Yuki Kanno.

**Investigation:** Yasuyuki Fujii, Ayano Hatori, Miwa Horiuchi, Risa Sugisaki, Yuki Kanno, Marika Sato.

**Supervision:** Tomoko Sugiyama-Tamura, Daichi Chikazu.

**Writing – original draft:** Yasuyuki Fujii, Ayano Hatori.

**Writing – review & editing:** Yasuyuki Fujii, Michihide Kono, On Hasegawa, Yoko Kawase-Koga, Daichi Chikazu.

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
