## [Decision Letter · Decision Letter 0]

10 Aug 2022

PONE-D-22-09899Computed tomography evaluation of risk factors for an undesirable buccal split during sagittal split ramus osteotomyPLOS ONE

Dear Dr. Yasuyuki Fujii,

Thank you for submitting your manuscript to PLOS ONE. After careful consideration, we feel that it has merit but does not fully meet PLOS ONE’s publication criteria as it currently stands. Therefore, we invite you to submit a revised version of the manuscript that addresses the points raised during the review process.

ACADEMIC EDITOR: It is indeed an interesting manuscript, however, I would suggest to answers the reviewers queries.

We look forward to receiving your revised manuscript.

Kind regards,

Tanay Chaubal

Academic Editor

PLOS ONE

Journal Requirements:

No competing interests exist.

Reviewers' comments:

Reviewer's Responses to Questions

**Comments to the Author**

1. Is the manuscript technically sound, and do the data support the conclusions?

Reviewer #1: Yes

Reviewer #2: Partly

2. Has the statistical analysis been performed appropriately and rigorously? 

Reviewer #1: Yes

Reviewer #2: I Don't Know

3. Have the authors made all data underlying the findings in their manuscript fully available?

Reviewer #1: Yes

Reviewer #2: Yes

4. Is the manuscript presented in an intelligible fashion and written in standard English?

Reviewer #1: Yes

Reviewer #2: Yes

5. Review Comments to the Author

Reviewer #1: 1) Nine surgeons who were trained in oral surgery for 9 or more years performed all operations:

no mention is made of intraoperative evaluation criteria (as supervisor has also been involved with 20 years of experience). Since huge number of surgeons are involved for the intervention (even though using the the same surgical technique with various surgical experience), there is high chances of inter-operater bias evaluating the outcomes. Therefore, authors should explain how they are going to minimise these bias. Moreover, surgeons to surgery ratio seems wider as 9 surgeons are intervening 53 ramus. This may be an important bias. Please try to minimise these shortcomings.

2) These measurements were conducted 3 times by 2 experienced oral surgeons, and the mean values were analyzed: how does the inter-observer bias during measurement process has been analyzed. what was the interval of measurement?

3)The age of the patients at the time of orthognathic surgery ranged from 17 to 51 years:

as the bone quality varies with increasing age, control group with 17 years and 51 years can show a potential error. how would you justify it? Furthermore, what was the indication for performing osteotomy for older aged groups?

4) No detail explanation about the surgical technique has been mentioned as this study is solely focused on intraoperative outcomes of osteotomy procedure.

5) Authors should stress the limitation of the study due to its nature and the fact that only one anatomical factor (thickness of distal bone of ramus) has been considered. This could influence surgical outcomes.

Reviewer #2: This study evaluated anatomical factors to cause a bad split in SSRO and concluded that bigger ratio of anterior to posterior width of mandibular ramus. There are some impediments to draw the conclusion. First of all, the limited number of cases. I am wondering if only 8 cases of bad split vs. 45 of successful split could perform sound statistical analysis. Furthermore, 9 surgeons operated though surgical technique might be one of the causes of bad split. As they stated in Discussion, the reason why the horizontal plane of mandibular foramen was selected to be analyzed is obscure. It might be better to analyze more sections. Lastly, I personally think the quality of bone such as elasticity could be critical factor of bad split. It might be interesting if they analyze CT value (HF unit) although I am not sure whether HF unit reflects the elasticity of bone.

6. PLOS authors have the option to publish the peer review history of their article (what does this mean?). If published, this will include your full peer review and any attached files.

Reviewer #1: **Yes: **Dr. Manoj Kumar Sah

Reviewer #2: **Yes: **Izumi Asahina

---

## [Author Response · Author response to Decision Letter 0]

14 Nov 2022

Point-by-point responses to the Reviewers’ comments

We wish to express our appreciation to the reviewers for their insightful comments, which have helped us to substantially improve our manuscript.

Responses to Reviewer 1

Comment #1

Nine surgeons who were trained in oral surgery for 9 or more years performed all operations:

no mention is made of intraoperative evaluation criteria (as supervisor has also been involved with 20 years of experience). Since huge number of surgeons are involved for the intervention (even though using the the same surgical technique with various surgical experience), there is high chances of inter-operater bias evaluating the outcomes. Therefore, authors should explain how they are going to minimise these bias. Moreover, surgeons to surgery ratio seems wider as 9 surgeons are intervening 53 ramus. This may be an important bias. Please try to minimise these shortcomings.

Response:

We greatly appreciate and agree with this comment. One expert in orthognathic surgery (the chairman of our department), participated and supervised all surgeries, to ensure the use of the same surgical technique and the same instruments. In addition, the other 8 surgeons, who were all certified surgeons of the Japanese Society of Oral and Maxillofacial Surgeons, have been performing oral and maxillofacial surgeries for 9 or more years. We hence assumed that all surgeons have basic skills and knowledge of orthognathic surgery, and performed all surgeries in the same way. In accordance with the comment, we have revised the text accordingly, and added this point as a limitation of the study (Materials and Methods: p5-6, lines 89-95, Discussion: p15, lines 263–269).

Comment #2

These measurements were conducted 3 times by 2 experienced oral surgeons, and the mean values were analyzed: how does the inter-observer bias during measurement process has been analyzed. what was the interval of measurement?

Response:

Thank you very much for this comment. To minimize inter-observer bias, the measurements were performed once a month, and 2 surgeons performed the measurements individually. In accordance with the comment, we have revised the text (Materials and Methods: p7, lines 125–126).

Comment #3

The age of the patients at the time of orthognathic surgery ranged from 17 to 51 years:

as the bone quality varies with increasing age, control group with 17 years and 51 years can show a potential error. how would you justify it? Furthermore, what was the indication for performing osteotomy for older aged groups?

Response:

We greatly appreciate and agree with this comment. As you mentioned, bone quality varies with age, and whether age is a risk factor for bad splits remains unclear (Br J Oral Maxillofac Surg. 2021 Jul;59(6):678-682. doi: 10.1016/j.bjoms.2020.08.107). In this study, there was no significant difference in the age of the patients between the control and the bad split groups, and hence we assumed that the bone quality of the patients in the 2 groups are similar. To confirm this point, we measured HU values to analyze bone quality (Fig 4. A-B), and our results showed that there were no significant differences between the 2 groups. However, posterior HUs were all higher than anterior HUs in the bad split group, and this feature might induce short splits in not only the lingual side but the buccal side. Further studies are needed to obtain more evidence regarding this point. We have included this information in the revised manuscript (Results: p10-11, lines 183–188).

Comment #4

No detail explanation about the surgical technique has been mentioned as this study is solely focused on intraoperative outcomes of osteotomy procedure.

Response:

Thank you very much for this comment. We have included a detailed description of the surgical technique to the text (Materials and Methods: p4-5, lines 76–88). 

Comment #5

Authors should stress the limitation of the study due to its nature and the fact that only one anatomical factor (thickness of distal bone of ramus) has been considered. This could influence surgical outcomes.

Response:

We greatly appreciate and agree with this comment. We only investigated 1 anatomical feature and bone quality in the ramus in this study. In accordance with the comment, we have added this limitation to the Discussion section (Discussion: p15, lines 254–259).

Responses to Reviewer 2

Comment #1

First of all, the limited number of cases. I am wondering if only 8 cases of bad split vs. 45 of successful split could perform sound statistical analysis.

Response:

We greatly appreciate and agree with this comment. In this study, patients who were treated during a 1-year period were analyzed, because some surgeons and instructors transfer to different departments or hospitals every year. In addition, surgical methods and instruments might differ among different operators. If we analyze patients who were treated within 2 or more years, we will be able to analyze more patients, but the data might include other biases or be more variable. Therefore, we analyzed patients who underwent surgery during a 1-year period. We agree with your comment that this is a limitation of this study, and have added this limitation to the manuscript. (Discussion: p15, lines 267–269).

Comment #2

Furthermore, 9 surgeons operated though surgical technique might be one of the causes of bad split.

Response:

Thank you very much for this comment. As you pointed out, differences in surgical experience or techniques might cause inter-operator bias in evaluating the outcomes. To minimize this bias, 1 expert of orthognathic surgery participated and supervised all of the surgeries, to ensure the use of the same surgical technique and the same instruments. In accordance with the comment, we included this information in the manuscript (Materials and Methods: p5-6, lines 89–95).

Comment #3

As they stated in Discussion, the reason why the horizontal plane of mandibular foramen was selected to be analyzed is obscure. It might be better to analyze more sections.

Response:

Thank you very much for this comment. In this study, we selected 2 areas at the height of the lingula of the mandible and the mandibular foramen, because these 2 points were easily identified in the horizontal image. It was sometimes difficult to perform measurements owing to metal artifacts, and to identify anatomical points for measurement in the other axial images. Further studies, such as the analysis of 3-dimensional images, are needed to obtain more evidence. In accordance with the comment, we added this information to the Discussion section of the revised manuscript (Discussion: p15, lines 254–262).

Comment #4

Lastly, I personally think the quality of bone such as elasticity could be critical factor of bad split. It might be interesting if they analyze CT value (HF unit) although I am not sure whether HF unit reflects the elasticity of bone.

Response:

We greatly appreciate and agree with this comment. In accordance with the comment, HU values in the ramus were analyzed to evaluate bone quality. The results indicated that there were no statistical differences in bone quality between the 2 groups. However, the anterior HU of the bad split group tended to be lower than that of the control group (p = 0.0717) (Fig 4C), and the values of posterior HU minus anterior HU in the bad split group also tended to be higher than in the control group (p = 0.0524) (Fig 4E). We hence could not conclude that there was a correlation between buccal splits and bone quality. However, posterior HUs were all higher than anterior HUs in the bad split group, and this feature might induce short splits in not only the lingual side but the buccal side. Further studies are needed to obtain more evidence regarding this point. In accordance with the comment, we added this point to the text (Results: p10-11, lines 183–188).

---

## [Decision Letter · Decision Letter 1]

6 Dec 2022

PONE-D-22-09899R1Computed tomography evaluation of risk factors for an undesirable buccal split during sagittal split ramus osteotomyPLOS ONE

Dear Dr. Yasuyuki Fujii,

Thank you for submitting your manuscript to PLOS ONE. After careful consideration, we feel that it has merit but does not fully meet PLOS ONE’s publication criteria as it currently stands. Therefore, we invite you to submit a revised version of the manuscript that addresses the points raised during the review process.

I thank you for replying to the reviewers comments. However, there are few minor queries which need to be addressed. Kindly reply to those queries.

We look forward to receiving your revised manuscript.

Kind regards,

Tanay Chaubal

Academic Editor

PLOS ONE

Journal Requirements:

Reviewers' comments:

Reviewer's Responses to Questions

**Comments to the Author**

1. If the authors have adequately addressed your comments raised in a previous round of review and you feel that this manuscript is now acceptable for publication, you may indicate that here to bypass the “Comments to the Author” section, enter your conflict of interest statement in the “Confidential to Editor” section, and submit your "Accept" recommendation.

Reviewer #1: (No Response)

Reviewer #3: (No Response)

2. Is the manuscript technically sound, and do the data support the conclusions?

Reviewer #1: Yes

Reviewer #3: Yes

3. Has the statistical analysis been performed appropriately and rigorously? 

Reviewer #1: Yes

Reviewer #3: Yes

4. Have the authors made all data underlying the findings in their manuscript fully available?

Reviewer #1: Yes

Reviewer #3: Yes

5. Is the manuscript presented in an intelligible fashion and written in standard English?

Reviewer #1: Yes

Reviewer #3: Yes

6. Review Comments to the Author

Reviewer #1: Most of the comments are addressed in an acceptable way.

Reviewer #3: The author has addressed most of the comments of the previous reviewers, However, there are other comments which are needed to be explained by the author: -

1- Could you define the bad split in the introduction section?

2- You have divided the patients into two groups, control and bad split, although they underwent the same procedures/split technique. Bad split is considered as a complication rather than a technique. I suggest renaming the groups such as correct/good split and bad split instead of control and bad split to be less confusing for the readers.

3- 9 Surgeons ran the surgery. These numbers of surgeons for 27 cases are too many and can increase the variability that can dramatically reduce your statistical power and impact the measurement and statistical data analysis, despite the presence of one supervisor. This point could be considered as a study limitation.

4- Could you mention the age of the patients in the methodology section?

5- You have mentioned that two surgeons measured the data. Is it for reliability or they shared the CT images to process them and collect data? If it was for interexaminer reliability, what is the correlation between the two measurements (surgeon 1 and 2)?

7. PLOS authors have the option to publish the peer review history of their article (what does this mean?). If published, this will include your full peer review and any attached files.

Reviewer #1: **Yes: **Manoj Kumar Sah

Reviewer #3: No

---

## [Author Response · Author response to Decision Letter 1]

13 Dec 2022

Point-by-point responses to the Reviewers’ comments

We wish to express our appreciation to the reviewers for their insightful comments, which have helped us to substantially improve our manuscript.

Responses to Reviewer 3

Comment #1

Could you define the bad split in the introduction section?

Response:

Thank you very much for this comment. We have revised the text in accordance with the comment (Introduction: p3, lines 44-45).

Comment #2

You have divided the patients into two groups, control and bad split, although they underwent the same procedures/split technique. Bad split is considered as a complication rather than a technique. I suggest renaming the groups such as correct/good split and bad split instead of control and bad split to be less confusing for the readers.

Response:

Thank you very much for this comment. To avoid confusion, we changed “control” to “good split” throughout the manuscript.

Comment #3

9 Surgeons ran the surgery. These numbers of surgeons for 27 cases are too many and can increase the variability that can dramatically reduce your statistical power and impact the measurement and statistical data analysis, despite the presence of one supervisor. This point could be considered as a study limitation.

Response:

We greatly appreciate and agree with this comment. In accordance with the comment, we have added this point as a limitation of the study (Discussion: p16, lines 273–275).

Comment #4

Could you mention the age of the patients in the methodology section?

Response:

Thank you very much for this comment. We have revised the text in accordance with the comment (Materials and Methods: p4, lines 69-71). 

Comment #5

You have mentioned that two surgeons measured the data. Is it for reliability or they shared the CT images to process them and collect data? If it was for interexaminer reliability, what is the correlation between the two measurements (surgeon 1 and 2)?

Response:

We greatly appreciate and agree with this comment. Two surgeons performed the measurements to examine interexaminer reliability. Interexaminer reliability was quantified using intraclass correlation coefficients (ICCs). The ICC values for the measurements of the CT images were from 0.966 to 0.998. The ICC value for the measurements of bone quality analysis in Fig. 4 was 0.972. These results indicated excellent reliability. We have revised the text accordingly, and added this point to the manuscript (Materials and Methods: p9-10, lines 165-167, Results: p11, lines 186–187, 193-194).

---

## [Decision Letter · Decision Letter 2]

15 Dec 2022

Computed tomography evaluation of risk factors for an undesirable buccal split during sagittal split ramus osteotomy

PONE-D-22-09899R2

Dear Dr. Yasuyuki Fujii,

We’re pleased to inform you that your manuscript has been judged scientifically suitable for publication and will be formally accepted for publication once it meets all outstanding technical requirements.

Kind regards,

Tanay Chaubal

Academic Editor

PLOS ONE

Additional Editor Comments (optional):

Reviewers' comments:

Reviewer's Responses to Questions

**Comments to the Author**

1. If the authors have adequately addressed your comments raised in a previous round of review and you feel that this manuscript is now acceptable for publication, you may indicate that here to bypass the “Comments to the Author” section, enter your conflict of interest statement in the “Confidential to Editor” section, and submit your "Accept" recommendation.

Reviewer #3: All comments have been addressed

2. Is the manuscript technically sound, and do the data support the conclusions?

Reviewer #3: Partly

3. Has the statistical analysis been performed appropriately and rigorously? 

Reviewer #3: Yes

4. Have the authors made all data underlying the findings in their manuscript fully available?

Reviewer #3: Yes

5. Is the manuscript presented in an intelligible fashion and written in standard English?

Reviewer #3: Yes

6. Review Comments to the Author

Reviewer #3: (No Response)

7. PLOS authors have the option to publish the peer review history of their article (what does this mean?). If published, this will include your full peer review and any attached files.

Reviewer #3: No

---

## [Editor Report · Acceptance letter]

15 Feb 2023

PONE-D-22-09899R2 

Computed tomography evaluation of risk factors for an undesirable buccal split during sagittal split ramus osteotomy 

Dear Dr. Fujii:

I'm pleased to inform you that your manuscript has been deemed suitable for publication in PLOS ONE. Congratulations! Your manuscript is now with our production department. 

Kind regards, 

on behalf of

Dr. Tanay Chaubal 

Academic Editor

PLOS ONE